# Vortex Target: A New Design for a Powder-in-Gas Target for Large-Scale Radionuclide Production

**Gerrie Lange**

GE Healthcare, Cygne Centre, De Rondom 8, 5612 AZ Eindhoven, The Netherlands; gerrie.lange@ge.com;
Tel.: +31-6-53796022

**Abstract:** This paper presents a design and working principle for a combined powder-in-gas target. The excellent surface-to-volume ratio of micrometer-sized powder particles injected into a forced carrier-gas-driven environment provides optimal beam power-induced heat relief. Finely dispersed powders can be controlled by a combined pump-driven inward-spiraling gas flow and a fan structure in the center. Known proton-induced nuclear reactions on isotopically enriched materials such as $^{68}$Zn and $^{100}$Mo were taken into account to be conceptually remodeled as a powder-in-gas target assembly, which was compared to thick target designs. The small irradiation chambers that were modeled in our studies for powdery 'thick' targets with a mass thickness (g/cm$^2$) comparable to $^{68}$Zn and $^{100}$Mo resulted in the need to load 2.5 and 12.6 grams of the isotopically enriched target material, respectively, into a convective 7-bar pressured helium cooling circuit for irradiation, with ion currents and entrance energies of 0.8 (13 MeV) and 2 mA (20 MeV), respectively. Current densities of ~2 μA/mm$^2$ (20 MeV), induces power loads of up to 4 kW/cm$^2$. Moreover, the design work showed that this powder-in-gas target concept could potentially be applied to other radionuclide production routes that involve powdery starting materials. Although the modeling work showed good convective heat relief expectations for micrometer-sized powder, more detailed mathematical investigation on the powder-in-gas target restrictions, electrostatic behavior, and erosion effects during irradiation are required for developing a real prototype assembly.

**Keywords:** cyclotron; powder target; thermal study; vortex target; Gallium-68; Technetium-99m

## 1. Introduction

Across the world, hundreds of cyclotrons with beam energies of 13 MeV and higher are applied for radionuclide production [1–3]. In the last decade, the cyclotron-based radionuclide production of $^{68}$Ga and $^{99m}$Tc has gained particular interest owing to the growing demand for $^{68}$Ga, and the expected shortages of the most widely used radionuclide, $^{99m}$Tc, obtained from $^{99}$Mo/$^{99m}$Tc generators. The cyclotron-based production of $^{99m}$Tc is an emerging technology that serves as an alternative to reactor-produced $^{99}$Mo. For the Netherlands (population 18 million), the total daily $^{99m}$Tc demand corresponds to 20 MeV proton irradiations of 12,000 μAh in 6-hour-run batches each day. This daily demand can be covered by one or two cyclotrons of 2 mA current. Growing demands for the radioisotope $^{68}$Ga for positron emission tomography can be met by an improved design of the $^{68}$Zn production target.

GE Healthcare uses an IBA Cyclone 30 cyclotron for the single-batch radionuclide production of over 1 TBq of $^{18}$F and $^{123}$I per day by irradiating isotopically enriched target materials [$^{18}$O]-water and $^{124}$Xe using beam currents of 180 μA and 300 μA, respectively [4,5].

A study at activated entrance windows by GE Healthcare and the University of Technology Eindhoven showed acceptable areal beam intensities up to 5 μA/mm$^2$. Correspondence with the cyclotron vendor confirmed that 2.0 milliamp cyclotrons are conceivable [5].

This study establishes a conceptual framework for a powder-in-gas target design with examples. Generally, beam power dissipation causes heat transfer challenges, and thus, the production capacity using a thick solid target is often limited by heat removal restrictions. The (technical) limitations are related to thermal properties, such as the target materials' conductivity and the heat transfer capabilities of the assembly. Prior to irradiation, preparation of target materials and the manufacturing processes of solid targets require pelletizing, sintering, and (multiple) Hydrogen gas reducing steps [1–3].

In this paper, the feasibility of the powder-in-gas target concept (vortex design) with a finely dispersed powder accumulated in an irradiation chamber is discussed.

## 2. Target Design

### 2.1. Vortex Target Design

The assembly design proposed here is based on an inert gas closed-loop circuit for removing the heat induced by the hitting beam to a secondary cooling water circuit. Micrometer-sized powder particles injected into such a gas circuit accumulate inside the circular arranged blade configuration, as indicated by the orange/red zone in Figure 1a. The purpose of the blades and the fan structure is to control the cylindrical-shaped area, where both the centrifugal and inward-directed drag and buoyant forces on the powder are balanced. The blade's front and end (Figure 1a) are conical and inwardly directed to establish small inward axial-directed particle drift. Thus, the conical sections prevent powder accumulation outside the orange/red zone. Figure 1b shows a radial cross-sectional view (A–A) half-way through the powder layer. The corresponding radial profile diagram indicates the angular and tangential gas velocities ($\Omega_{gas}$ and $v_{g.tan}$). The tangential gas velocity $v_{g.tan}$ increased from radial point 1 to a maximum indicated value at radial point 3, which is close to the fin tips of the elongated centered fan structure. The enforced gas spinning (region inside radial points 2 and 3, Figure 1b) and subsequently enhanced centrifugal force lead to the continued accumulation of powder particles in the indicated cylindrical orange/red zone with a length denoted by $L_{layer}$. The product of the dispersed powder density and length $L_{layer}$ is equal to the powder-in-gas mass thickness (i.e., g/cm$^2$) and must correspond to the thick target values.

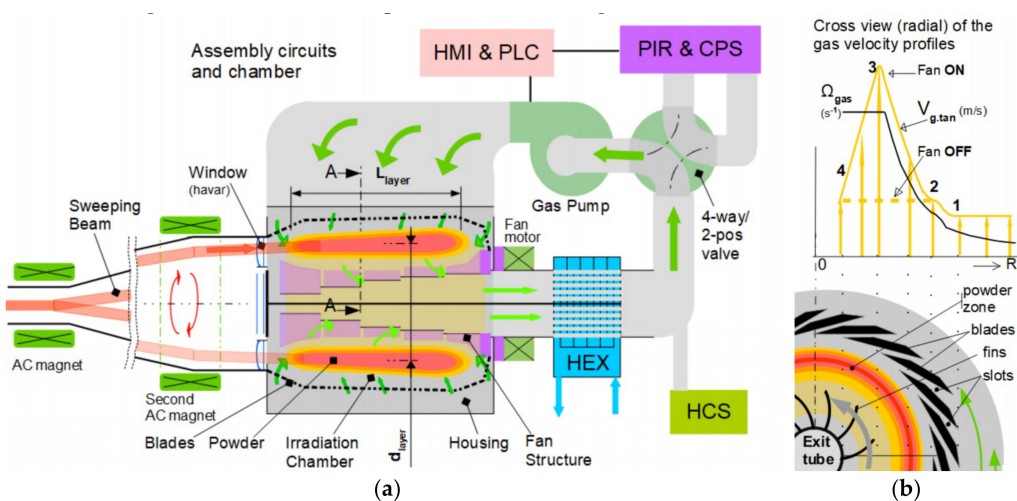

**Figure 1.** (**a**) Sketch of the gas circulation system setup with the beam guiding system, including a dual AC magnet configuration. The volume region in which the powder accumulates is shown in the irradiation chamber. (**b**) Cross-section (A–A) and diagram of the angular and tangential velocities of the gas rotation vs. radius. The numbers in the diagram indicate the velocity profile changes versus radius.

Figure 1a shows the gas circuit components and equipment for process control, including the central parts: cochlea (housing); window section; irradiation chamber with the blades' configuration; and elongated fan structure, which is driven by a magnet-coupled fan motor. The gas pump, shown in

green in Figure 1a, generates an inert gas flow that passes the blades and spirals strongly inward to the centered exit tube inside the fan structure.

Other equipment labeled in the figure include:

- Heat exchanger (HEX) for heat removal by the secondary cooling water circuit;
- Helium control system (HCS) for the regulation of the circuit pressure and gas temperatures;
- Four-way/two-position valve for loading and emptying the irradiation chamber;
- Powder injection and recovery (PIR) system and Chemical Processing System (CPS); and
- Process control (PLC) and operator panel (HMI).

Prior to the operation, residual gasses in the target chamber are evacuated. Subsequently, the chamber and circuit are helium pressurized, and the gas pump and the central fan structure are turned on. As a result of the 4w-valve operation, the injected powder is dispersed into the circuit and accumulates in the (indicated orange/red) cylindrical zone.

Subsequently after irradiation, the powder circulating in the irradiation chamber can be scavenged in the PIR system using the 4w-valve operation and reduction of the fan's spinning structure while the gas pump continues inert gas circulation. Reducing the fan's spinning frequency leads to a reduction in centrifugal forces operating on the powder's particles, resulting in a further inward and exiting powder transfer. The decrease in the tangential gas velocity is indicated in Figure 1b by the shift of point 3 to the dashed line level.

## 2.2. Principle of the Vortex

When the operation begins, the powder is dispersed and brought by the carrier gas into the indicated orange/red zone. In this zone, a balance of all forces in the radial direction must be achieved. The carrier gas rotation induces a centrifugal, a buoyant, and a drag force on the particles. The drag force is related to the particle's drift velocity relative to the gas.

For the particles present in the balanced zone, as illustrated in Figure 2, radial forces are expressed by Equation (1):

$$F_{cen} = F_{drag} + F_{buo} \tag{1}$$

where $F_{cen}$ is the centrifugal force, $F_{drag}$ is the drag force, and $F_{buo}$ is the buoyant force.

The main expression for $F_{cen}$ is:

$$F_{cen} = \frac{2 \cdot m_p \cdot v_{g.tan}{}^2}{d_{layer}} \tag{2}$$

where $m_p$ is the particle's mass (kg), $d_{layer}$ is the average powder layer diameter, and $v_{g.tan}$ is the entering gas velocity equal to the quotient of gas volume flow rate and cross-section of gas inlet.

Calculating drag force $F_{drag}$—on the expected micrometer-sized particles moving in viscous gas—can be described by Stokes' law, which is accurate in a gaseous environment with a Reynolds number of Re ≤ 0.1. For particles having Reynolds numbers of Re ≤ 1.0, Stokes' law remains a proper approximation [6]. Preliminary calculations showed that the range of interest for the particle's size was smaller than 10 μm, while the differential or relative velocities to the carrier gas were expected to be ~1.0 m/s. The Reynolds number verification was carried out for circulating helium gas in the irradiation chamber at a gas density of $\rho_g$ = 1.25 kg/m$^3$ ($\approx$ 7E + 05 Pa, 300 K) and a dynamic viscosity of μ = 2.1E − 05 Pa·s. Particle calculations in the expected ranges of size and velocity ($d_p$ < 10 μm, $v_{p.rel} \approx$ 1.0 m/s) by Equation (3),

$$Re_p = \frac{\rho_g \cdot d_p \cdot v_{p.rel}}{\mu}, \tag{3}$$

resulted in Reynolds numbers of $Re_p$ < 0.5. Herein, $\mu$ is the dynamic viscosity of the carrier gas, $d_p$ is the particle diameter (m), $\rho_g$ is the gas density (kg/m$^3$), and $v_{p.rel}$ is the differential velocity of the particles relative to the gas. The dynamic viscosity's temperature dependency was investigated and determined to be of minor significance for this study.

The drag formula for low differential velocities is expressed by:

$$F_{drag} = 3 \cdot \pi \cdot \mu \cdot f_{eff} \cdot d_p \cdot v_{p.rel} \tag{4}$$

where $f_{eff}$ is a factor for irregular particle surface condition.

The next equation shows the balance of buoyant force $F_{buo}$ and drag force $F_{drag}$ equal to the centrifugal force $F_{cen}$ by:

$$\frac{\pi \cdot d_p{}^3 \cdot \rho_g \cdot v_{g.tan}{}^2}{3 \cdot d_{layer}} + 3 \cdot \pi \cdot \mu \cdot f_{eff} \cdot d_p \cdot v_{p.rel} = \frac{\pi \cdot d_p{}^3 \cdot \rho_p \cdot f_p \cdot v_{g.tan}{}^2}{3 \cdot d_{layer}} \tag{5}$$

where $d_{layer}$ is the average diameter of the intended powder zone inside the blades, and $v_{g.tan}$ is the tangential gas velocity. The expressions $f_p$ and $f_{eff}$ are correction factors for the particle's density and surface roughness, respectively. The variables $\rho_g$ and $\rho_p$ are the densities (kg/m$^3$) of gas and particles, respectively.

The extraction of the particle's velocity relative to the gas results in Equation (6):

$$v_{p.rel} = \frac{d_p{}^2 \cdot v_{g.tan}{}^2 \cdot \left(\rho_p \cdot f_p - \rho_g\right)}{9 \cdot \mu \cdot f_{eff} \cdot d_{dlayer}} \tag{6}$$

Of course, Equation (6) can be used for areas other than the balanced zone by redefining the quantity $d_{layer}$ by a new expression for the diameter or twice the radius.

Particles not exceeding a certain size or diameter will be transferred inward by the carrier gas, as indicated by the small brown radial resulting velocity vector $v_{p.rad.res}$ (Figure 2, #1).

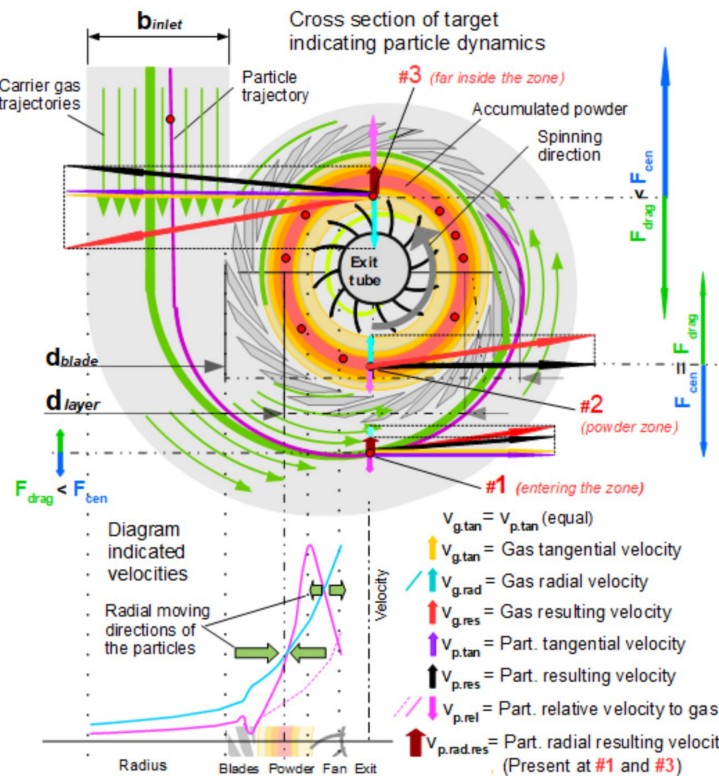

**Figure 2.** Illustrations of the gas trajectories by (curved) green arrows of equal lengths in the housing. The three positions labeled #1, #2, and #3 describe the locations that are outside, inside, and "too far" inside the powder presence zone. The diagram shows that the radius depends on velocity trends for the gas and the particles. Obviously, the particle relative velocity (pink line) rises beyond the radial gas velocity (blue line).

The inward-spiraling carrier gas has a radial velocity component $v_{g.rad}$ (light blue arrow) that is related to gas velocity $v_{g.tan}$, quotient of the indicated housing dimensions $b_{inlet}$, and the circumference of the blade's configuration in the cochlea (Figure 2, #1). Position #2 indicates the location of equal forces without the presence of a brown radial resulting velocity vector $v_{p.rad.res}$. Enhanced gas spinning far inside the zone (position #3) results in a velocity vector $v_{p.rad.res}$ directed outward. Green block arrows in the diagram show the particles' radial moving directions.

Density of the powder material is related to its porosity. Therefore, the factor $f_p$ is defined and estimated to be between 0.5 and 0.9. The shape and (irregular) surface finish correction factor $f_{eff}$ represents the multiplying factor for the diameter $d_p$. The factor $f_{eff}$ varies from 1.0 for a smooth surface to higher values for irregular surfaces. The size, density, and surface finish are surely affected by the preparation of the powder. A preparation procedure must be developed to determine the optimal range of the powder's size that can be applied for injection.

The purpose of carrier gas circuits is to transfer the dissipated heat induced by the beam outside the target system. When the maximum allowed temperature rise, $\Delta T_{gas}$, and circuit pressure of the carrier gas are defined for an expected beam power load $\dot{Q}_{tot}$ (Watt), the mass flow rate $\dot{m}_g$ (kg/s) and volume flow rate $\dot{V}_{fl}$ (m³/s) of the gas cooling can be calculated.

*2.3. Stopping Power, Ranges, and Beam Features*

To model the system, two radionuclide production routes, $^{68}$Zn(p,n)$^{68}$Ga and $^{100}$Mo(p,2n)$^{99m}$Tc, were taken into consideration. For determining the design, the significant factors are the accelerated ion energy at the entrance, the projected ranges related to the electronic and nuclear (minor) stopping powers, and the lower threshold energy for the considered nuclear reaction [7,8].

The ion beam, which enters the assembly nearly parallel to the central assembly's symmetry axis, is intended to pass the full length $L_{layer}$ of the mixed powder-in-gas layer. The powder-in-gas mass thickness (i.e., g/cm²) should correspond closely to the values of known thick targets. The rest of the energy from the ion beam, assumed to be less than the specific nuclear reaction threshold energy, dissipates at the end section of the blades. The ion energy losses due to scattering at the entrance window, the carrier gas, mixed powder-in-gas layer, and blades are defined by Equation (7):

$$\mathrm{E}_{ion} = \Delta\mathrm{E}_{havar} + \Delta\mathrm{E}_{gas.1} + (\Delta\mathrm{E}_{mat} + \Delta\mathrm{E}_{gas.2})_{mixed} + \Delta\mathrm{E}_{blade} \tag{7}$$

where $\Delta\mathrm{E}_{havar}$ and $\Delta\mathrm{E}_{gas.1}$ are the energy losses in the window and the first section of the carrier gas, respectively. Generally, beam scattering and energy loss $\Delta\mathrm{E}_{gas.1}$ in the carrier gas are expected to be minor. The expression $(\Delta\mathrm{E}_{mat} + \Delta\mathrm{E}_{gas.2})_{mixed}$ is the ion beam energy loss due to both dispersed powder $\Delta\mathrm{E}_{mat}$ and carrier gas $\Delta\mathrm{E}_{gas.2}$ in the same volumetric region. The relative contribution of the latter is much lower than the former.

For a technical assessment of the concept, the maximum temperature rise of the powder particles was estimated. Temperature rise depends on the energy level of the local proton beam hitting the particles. For a particle at the start of the layer near the target's entrance, energy loss will be significantly lower than that for a particle at the end of the passed powder-in-gas layer ($L_{layer}$). Otherwise, beam intensity (µA/mm²) at the entrance is significantly higher compared to when it is further "away" inside the irradiation chamber. To account for worst-case scenario, we calculated the temperature rise of a cylindrical-shaped particle (Figure 3b), with an energy loss at the maximum stopping power at the top of the Bragg peak. Therefore, the beam intensity was calculated at the end of the powder-in-gas layer and supported by ion range and scattering (SRIM) calculations [7–9].

A particle's passage (in a static beam) driven by the tangential velocity $v_{g.tan}$ occurs in a few milliseconds, while particle heating occurs instantly in tens of microseconds. The heating and convective cooling of a particle reach equilibrium at a differential temperature $\Delta T_{tr}$ relative to the gas. The maximum dissipated ion energy $\Delta\mathrm{E}_{max}$ in MeV per powder particle (1 eV = 1.602E − 19 J), with the diameter $d_p$ and density $\rho_p$, is by approximation:

$$\Delta E_{max} = SP_{max} \cdot \rho_p \cdot d_p \tag{8}$$

where $SP_{max}$ is the maximum mass stopping power at the Bragg peak in MeV·cm$^2$/gr. The dissipated beam power $\dot{Q}_p$ in a powder particle is:

$$\dot{Q}_p = \pi/4 \cdot d_p{}^2 \cdot I_{int.static} \cdot \Delta E_{max} \tag{9}$$

where $I_{int.static}$ is the beam intensity in µA/mm$^2$ (or µC/(s·mm$^2$)), which corresponds to the accelerated ion particle's 'flow rate' per square millimeter.

The maximum powder particle differential temperature $\Delta T_{tr.staic}$ relative to the gas is then:

$$\Delta T_{tr.static} = \frac{\dot{Q}_p}{h_{He} \cdot A_p/2} \tag{10}$$

where $A_p$ is the cylindrical particle surface divided by two, given the assumption that only the front-half of the particle's surface is cooled.

Otherwise, when the maximum for the differential temperature $\Delta T_{tr.max}$ is set for the worst-case scenario of energy loss due to the mass stopping power $SP_{max}$ at the "Bragg peak" for a known particle diameter $d_p$ and density $\rho_p$, the maximum allowed beam intensity can be calculated by the following formula:

$$I_{int.static.max} = \frac{3 \cdot h_{He} \cdot \Delta T_{tr.max}}{d_p \cdot \rho_p \cdot SP_{max}} \tag{11}$$

where $I_{int.static.max}$ is the maximum for the static flat-top beam profile, and $h_{He}$ is the heat transfer coefficient (W/m$^2$·K), which is determined by gas flow data and explained in the discussion.

The flat-top beam profile reduces the damaging effects of hotspots on the (2 × 15 µm Havar) windows and allows a higher beam power while keeping the maximum allowed peak current density noted in the introduction (5 µA/mm$^2$) unchanged. Further decrease in beam intensity or a higher allowed total beam power can be established by sweeping the flat-top beam around the assembly's symmetrical center. Sweeping around the center further reduces the window's heat stress as well as the particle heating. Preliminary calculations showed that due to the instantaneous heating of powder particles, a beam sweeping frequency of 1 kHz results in a significant reduction in the particle's differential temperature $\Delta T_{tr}$. Figure 3a shows an impression:

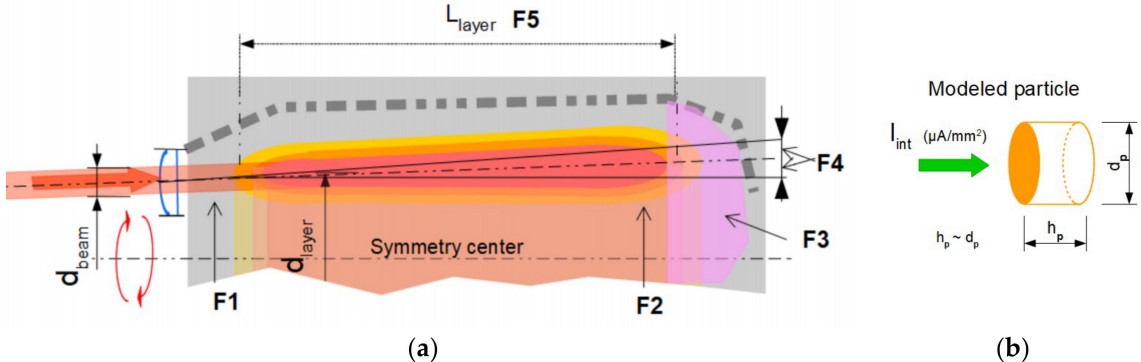

(a)                                   (b)

**Figure 3.** (**a**) Cross-sectional view of the powder layer in the irradiation chamber, where F1 is the entering beam at an energy of 14–30 MeV; F2 is the expected beam area at a threshold energy of 4–10 MeV; F3 is the area with the highest stopping power (Bragg Peak); F4 is the scattering-induced lateral range and straggling; and F5 is the length of the powder layer corresponding to thick targets setups. (**b**) Modeled particle.

## 3. Results

To optimize the vortex target design, an iterative modeling process using ion range and scattering (SRIM) data as well as heat transfer calculations was carried out for two nuclear reaction routes [8–10].

Initial data processing involved determining (in the order presented): beam features, ion range and scattering, and particle features and specifications.

Next, particle velocity calculations were performed for the numbered positions indicated in Figure 2 of the following:

- Position #1: the particle in the entering zone and confirmation of the inward-directed transfer;
- Position #2: the particle in the powder zone and confirmation of the balanced particle presence; and
- Position #3: calculation of 'too far' inside the powder zone and confirmation of the outward drifting of the particles.

Confirmation by approximated calculations of particle heating and convective heat relief.

Table 1 shows the input and calculated results of the modeling work of a high-capacity $^{68}$Zn and $^{100}$Mo powder-in-gas target. The beam enters the target chamber almost horizontally in a sweeping mode by the beam guiding system. Beam energy loss due to interaction with the powder layer over the full length is equal to the difference between $E_{ion}$ and $E_{threshold}$. Particle size in the table indicates a certain accuracy range for the operation. Particles outside this range will be transferred to the PIR unit. Total powder mass for 'thick targets' depends on the flat-top ion beam size $d_{beam}$, the nominal diameter of the powder layer $d_{layer}$, powder's density, and beam scatter (Figure 3a).

**Table 1.** Input data and results of $^{68}$Zn and $^{100}$Mo modeling and calculations.

| | Input data and results | Quantities | Unit | $^{68}$Zn(p,n)$^{68}$Ga | $^{100}$Mo(p,2n)$^{99m}$Tc | Remarks |
|---|---|---|---|---|---|---|
| | Proton Beam energy | $E_{ion}$ | MeV | 13 | 20 | |
| | Beam current | $I_{tar}$ | mA | 0.8 | 2.0 | |
| | Threshold energy | $E_{threshold}$ | MeV | 3.8 | 7.8 | |
| | Particle size | $d_p$ | μm | 4 ± 1.5 | 3 ± 1 | |
| | Target thickness | $L_{matter}$ | mm | 0.38 | 0.60 | |
| | Mass thickness (average) | Mass thickness | gr/cm$^2$ | 0.27 | 0.62 | |
| | Powder layer length | $L_{layer}$ | mm | 40 | 60 | |
| | Diameter powder zone | $d_{layer}$ | mm | 42 | 56 | |
| | Beam areal intensity | $I_{sweep}$ | μA/mm$^2$ | 1.35 | 1.81 | |
| | Powder's total mass | $m_{matter}$ | gr | 2.5 | 12.6 | |
| #1 | Particle relative velocity | $v_{p.rel\,(#1)}$ | m/s | 0.44 | 0.44 | |
| | Gas radial velocity | $v_{g.rad\,(#1)}$ | m/s | 1.69 | 1.82 | |
| | Particle incidence angle | $\psi_{fl\,(#1)}$ | deg | 8.06 | 6.87 | (>0 = Ok) |
| #2 | Particle relative velocity | $v_{p.rel\,(#2)}$ | m/s | 2.70 | 2.93 | |
| | Gas radial velocity | $v_{g.rad\,(#2)}$ | m/s | 2.71 | 2.91 | |
| | Particle incidence angle | $\psi_{fl\,(#2)}$ | deg | −0.00 | −0.03 | (≈0 = Ok) |
| #3 | Particle relative velocity | $v_{p.rel\,(#3)}$ | m/s | 8.45 | 10.30 | |
| | Gas radial velocity | $v_{g.rad\,(#3)}$ | m/s | 4.06 | 4.76 | |
| | Particle incidence angle | $\psi_{fl\,(#3)}$ | deg | −9.02 | −7.61 | (<0 = Ok) |
| | Diff. temp. static beam | $\Delta T_{tr.static}$ | K | 624 | 670 | Both not allowed |
| | Diff. temp. sweeping beam | $\Delta T_{tr.sweep}$ | K | 270 | 300 | Freq. 1 kHz |
| | Max. sweeping beam intensity | $I_{int.sweep.max}$ | μA/mm$^2$ | 2.13 | 2.05 | |

The differential temperatures $\Delta T_{tr.static}$ and $\Delta T_{tr.sweep}$ (1 kHz sweeping beam) were calculated at the maximum mass stopping power (Bragg Peak). The maximum sweeping beam intensity $I_{int.sweep.max}$ at the end of the powder layer was calculated for a scattered beam on a modeled cylindrical particle (Figure 3b). The values for the differential temperature $\Delta T_{tr.static}$ are interpreted as not acceptable.

## 4. Discussion

The design work showed an interesting route toward achieving a powder-in-gas vortex target. However, several processes, such as preventing adverse powder accumulation outside the intended

layer, must be investigated in detail. The presence of the fan structure is essential for supporting the layer's stability by the carrier gas's enhanced tangential velocity in the irradiation chamber.

Generally, prior chemical processing results in different shapes, sizes, and surface conditions of particles. An apparatus, possibly identical to a vortex assembly, must be developed for powder particle size selection, and reprocessing powder particles that exceed the desired 'range'.

The heat transfer coefficient $h_{He}$ and gas flow velocity $v_{He}$ at the contact surface were examined in this study using the pressure and gas flow data of existing and designed assemblies. The heat transfer coefficient has close to a linear dependency on the circuit pressure $P_{cir}$ and gas velocity $v_{He}$. An empirical expression $\xi_{He}$ was determined and is shown in the following formula:

$$h_{He} = \xi_{He} \cdot P_{cir} \cdot v_{He} \tag{12}$$

where $\xi_{He}$ is defined as the 'heat transfer constant' $\xi_{He}$ between $1 \times 10^{-4}$ and $4 \times 10^{-4}$ W·s/(Pa·K·m$^3$). Further investigation of the basic heat transfer coefficient is recommended to determine helium flow angle dependency relative to the (particle's) surface.

Each powder has a certain level of hardness. This potentially induces erosion effects on the blades and other components. Candidate materials for internal structures must have excellent properties for thermal and electrical conductivity, negligible long-term radio-activation profiles, and significant chemical differences from the produced radionuclide.

When a beam passes the target containing the powder and carrier gas, they become partly ionized. The gas will contain a dilute plasma of free electrons and positive ions determined by a balance between beam-induced ionization and recombination processes. The powder particles become positively charged primarily by secondary electron emission induced by the impinging accelerated ions. The target's design is intended to cause particles, when attracted to the blades, to discharge and re-enter the powder layer by the intensive tangential gas flow. Further research into the particle's behavior during irradiation conditions is addressed in Section 5. Further, quantities such as the net ion current, as well as differential gas circuit pressures and various temperature positions in the assembly's structure, must be monitored.

Our calculations showed that the instantaneous heating of particles occurs in tens of microseconds during beam passage. The average differential temperatures, as shown in Table 1, were calculated for a beam sweeping frequency of 1 kHz. When the beam sweeps in the opposite direction relative to the powder rotation, the particles' beam passage, heating, and cooling occur in tens of microseconds. Further increasing this frequency leads to more averaging of particle heating, resulting in reduced temperature of individual particles.

## 5. Conclusions

This study suggests that the excellent surface-to-volume ratio of micrometer-sized particles in a carrier-gas-driven environment leads to optimal heat relief. Gas flow induced by the gas pump transfers the powder inside the blade's structure. Inside, the fan's structure maintains the accumulated powder near the blade's structure via enhanced tangential gas velocity.

The advantages of the powder-in-gas target design are:

- Shortened production cycles when higher beam intensities are applied;
- Expected shortened target material preparation procedures; and
- Faster recovery, dilution procedures, and reprocessing to powdery material.

Assemblies for nuclear reactions on $^{68}$Zn and $^{100}$Mo materials were modeled as a powder-in-gas target. The conceptual design of small-diameter chambers for $^{68}$Zn and $^{100}$Mo revealed the need for 2.5 and 12.6 grams of material, respectively, for a beam current operation of 0.8 and 2 mA, respectively. Calculations were carried out for the worst-case scenario of particle heating. Beam current densities were calculated close to 2 μA/mm$^2$ and corresponded to energy-dependent power loads of ~4 kW/cm$^2$.

Further detailed examinations are required to:

- Determine the optimal cochlea, blade shape, and fan structures;
- Reduce and control erosion effects;
- Optimize the preparation of powder and its injection into the target system;
- Determine the tendency of powder to form dendrites or small deposits at the blades;
- Avoid agglomeration of sponged particle formation due to discharging effects;
- Control particle drift to the blade's surfaces by, for instance, (proton) current measurement; and
- Determine the effect of applying a (positive) electrical voltage to the fan structure and control powder's behavior possibilities during irradiation.

Summarizing the conceptual design, the nuclear reactions: $^{68}Zn(p,n)^{68}Ga$ and $^{99}Mo(p,2n)^{99m}Tc$, could be good candidates for large-scale production because of the broad interest and their use across the world. This study shows that the concept might be also applicable to other production routes.

**Funding:** This research received no external funding.

**Conflicts of Interest:** The author declares no conflict of interest.

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
