# Peer review of "Vortex Target: A New Design for a Powder-in-Gas Target for Large-Scale Radionuclide Production"

_instruments, doi:10.3390/instruments3020024_

Round 1

Reviewer 1 Report

The paper and the content is interesting and could be a promize for large scale radionuclides production.

It is explained in detail and in a very good way all the mathematic part that justify the physics processes, specially for the heat issue.

The heat transfer issue is well analyzed and the modelling and the calculations indicate a promising outcome.

- But I think that is missing all the part related to the quantification of the radionuclides production.

- I think that this system could be used in general for any kind of radionuclide. Why do you cite in particular the production of 44-Sc, 68-Ga, 99m-Tc, 99-Mo?

- for me it is not clear the amount that you can produce compared with the Thick Target Yield (MBq/C) of, for example, metalic targets.

- did you take into account the contamination, the side nuclear rections or the nuclear reactions in the chamber?

Few corrections:

- in Tab.1 Gray shaded cells ... - the grenn shaded cells ...

- the same at line 318

- all the elements must be un lower case when are written not as symbol; like zirconium, helium and so on

- line 131: In appedix A, both figures 7a and 7b ... and no 10a and 10b ...

- line 132 the same as line 131: 7c and 7d ... and no 10c and 10d

- line 154 The purpose of the carrier gas is to transfer the heat of the system. Is it "transfer" to or where? or "remove?

- line 155 the meaning of ' in  μA's and Mev's

- line 188: ... for the nuclear reactions [without ()]

- line 251; is calculated and found  ....

Table 3.in column 2 Quantities, second line: E prot or Epart?

line 383: of the beam energy. (delete . before beam)

Author Response

Dear reviewer

Thank very much for the reviewing work

All the corrections are processed, but the other reviewers requested a shorter version of the article and minor presence of technically related data and formulas.

The responses to the comments are attached.

Thank you so far.

Gerrie Lange

Reviewer 2 Report

General comments:

General comments:

The manuscript reports an, even too much detailed, technical description about a concept design study for a combined powder-in-gas target for a large-scale radioisotope production, such as Mo997Tc99m, Ga68 and the theranostic Sc44. The proposed approach basically exploits the dispersion of micrometre-sized particles in a forced carrier gas, which turbulent spiralling flow pattern and steady state conditions may be controlled inside a proper gas chamber. The idea to have a gas/liquid carrier for transporting a fine powder of isotopic-enriched target material is not new; however such approach in the past has never produced interesting results.  

I have read the manuscript and found it unusually too much long and confusing, even for the well-trained reader in the subject. The subject is not complicated; however, this is not a scientific paper. This manuscript instead appears to be a cut-and-paste operation of a technical design report. Too much useless technical details/parameters (including 3 Appendices at the end to further complicate the already too long description, as well) have indeed been reported. The main focus of this work is the main cinematic/dynamic parameters describing the behaviour of the spiralling carrier He gas approaching the steady-state in a narrow ring-shaped cylindrical region where the accumulation of the isotope-enriched powder is supposed to occur. The same author at last concludes that, despite the promising design some parameters have been investigated more on detail. At the end, it is not clear if a prototype of the system concerned has ever been constructed. 

On the other hand, no any estimations about what would have been much more interesting to report, i.e. the yield expected, in REAL irradiation conditions (taking into account the maximum beam currents available from existing cyclotrons) is reported for the radioisotopes concerned. That might help the reader to better understand the technical potential of such a target system. Moreover, all the issues related to the control of all the radioactivity of powders, freely circulating inside the target chamber, are never reported. Anyway one of the main limiting issue I see for such a system is the tremendous amount of isotopic-enriched powder mass required, (several tens/hundreds of grams which is dramatically higher than the mass required by the other more traditional solid target concepts. 

Specific comments

Quite a deep revision (basically a brand-new, more scientific and balanced manuscript, not longer than the half of current manuscript) is needed prior it might be accepted for publication. The revisions suggested, as well as specific comments, are reported directly on the Pdf file of the manuscript attached, using the commenting tools. They have to be considered by the authors as a reference for requested revision in manuscript drafting. Authors are asked to improve the manuscript in the highlighted parts and to meet the revisions requested or to make, in addition, their own modifications.

Author Response

Dear reviewer

Thank for all the valuable comments.

The article is completely revised with strong focus to the basic approach, namely, the particle dynamics and heat relief in a powder-in-gas target.

The article is shortened, the excel sheet is removed.

some technical data with other explanation info is added in a Supplementary Information document.

The response to the comments is attached.

Gerrie Lange

Reviewer 3 Report

This is an extremely busy paper making a lot of assumptions with virtually no experimental evidence. The figures are very complex as is the text.

There author is encouraged to think about simplifying the manuscript and highlighting only the very important issues.

While the concept is interesting and the author makes some good assumptions there are some serious flaws in the reasoning that will be made clear by experimental investigation. For example, the fact that Nb is ductile makes it MORE prone to abrasion and deformation, therefore not good as a blade material.

While this reviewer thinks the author should NOT make assumptions of 1-10mA of beam current, I do think the work can be published as an academic exercise. Please make a serious effort to fix the english.

Author Response

Dear reviewer

Thank for all the valuable comments.

The article is revised significantly and shortened.

Most technical data is removed or moved to a new Supplementary Information document.

The response to the comments is attached.

Gerrie Lange

Round 2

Reviewer 1 Report

In this form is better.

For me, remain the confusion for the reader why you report the production of Tc-99m, Ga-68 expecially in the abstract.

You don't report an experimental part with a comparison with the data of, for example, Thick Target Yield. You have to demonstrate that your system is better to the traditional way to produce the radionuclides.

I know that this is not the focus of your paper, but you write "Known proton induced nuclear reactions on the isotopically enriched materials Zinc-68 and Molybdenum-100, are conceptually remodeled as powder-in-gas target assembly and compared to thick targets designs. Production of Technetium-99m using powder-in-gas target assemblies by currents 5 mA or more, might be able to compete economically with nuclear reactor-based production of Molybdenum-99. "

For example, you can say at 2.4 that as an example in the future for the properties [but you have to add some considerations/properties or ....] the nuclear reactions as 68Zn ... and 100Mo could be a good candidates for ..... because ....

Moeover you never explain if the chamber will be contaminated at the ned of the irradiation and so how do you plan to handle this situation also from radioprotection point of view; how many hours of irradiation to obtain the same amount of activity of the radionucledes that you cite.

So I suggest that you say clearly that all the pat related to the real production will be presented in the future when all the system will run and you will have data to really compare.

Remember that the name of the element MUST be in lower case; as molibdemum-100. Otherwise 100Mo or Mo-100, in capital letter. So check all the text.

The same liter or L (no Liter).

Author Response

Dear reviewer

Thank you for the comments and suggestions.

I have revised the document strongly. 

The document is language edited by the MDPI services.

The document is changed into a Communication Document instead of an article. 

The design is changed in a powdery thick target set-up in the front with a nearly horizontal entering beam direction. The powder's mass required is hereby very small. The comparison and other claims are removed. Modeled and calculated data are presented now. 

The comments;

Point 1: The report of Tc-99m and Ga-68 in the abstract.

Response 1: These radionuclides are used as modeling as examples due to the range of interest across the world. The powder-in-gas concept might be useful for other production routes as well.

Point 2: Don't report experimental data. Demonstrate that system is better.

Response 2: Due to the redesign of the chamber and beam direction, the design is much more obvious. The modeling work is established for ion energy ranges at the entrance to the threshold energy for specific nuclear reaction. The length of the powder layer corresponds to thick target in the model and calculations.

Point 3--4: The 5 mA claim regarding the competition with reactor based Mo-99 and the nuclear reactions for Ga-68 and Tc-99m could be good candidates.

Response 3--4: The comparison claims are removed. However, focusing on the Netherlands, about 5 to 6 cyclotrons would be needed using 300 μA beams. The report showed special attention to these nuclear reactions due to the broad range of interest and use. 

Point 5: The contamination of the chamber and radioactive protection.

Response 5: For the Netherlands, one or two cyclotrons running 2 mA daily 6 hours would cover the needs. Handling activation and contamination has to be controlled but every irradiation assembly becomes activated. The choice of materials and design is important. Comparing with our existing site at Eindhoven (Cygne center), daily production of multiple 1 TBq batches of I-123 and  F-18 on routine basis is possible by good material choices and almost seal-less assemblies.

Point 6: Future presentation of real experimental data.

Response 6: This Communication Document is related to the WTTC17 in Portugal. Experiments in near future requires resources which are not yet available. Conventional experimental work regarding vortex is foreseen before the WTTC18 meeting.   

Point 7--8: Molybdenum and Liter.

Response 7--8: Document is checked and language edited by the mdpi services.

Thanks again,

Gerrie Lange

Reviewer 2 Report

General comments:

The manuscript new version submitted by the author is now much more balanced. However, even with such a revised work, I remain doubtful about the effectiveness of such a powder-in-gas target system, with respect to the current solid target ones. The main focus of this work still remains the cinematics/dynamics parameters determination describing the behaviour of the spiralling carrier He gas approaching the steady-state, in a narrow ring-shaped cylindrical region where the accumulation of the isotope-enriched powder is supposed to occur. The same author at last concludes that, despite the promising design, some parameters have still to be investigated more in in depth. The main motivation given by the author is always the feasibility study about the very high heat power relief (in my opinion untrustworthy values) the syste would be able to manage, as also reported in Table 1 (e.g. the excessively high 100 kW beam power – at Which energy?) for Mo100(p,2n)Tc99m reaction, that I consider basically unpractical.    

On the other hand, the information I did expected, about the resulted target mass thickness that such a system is able to manage, is still missing (e.g. for Mo100-enriched powders). Such a parameters is critical to better understand how, such a powder-in-gas target system, is far from a standard thick solid target (e.g. for Tc99m production). Therefore how much activity per unit mass and electric charge of a given radiosiotope (e.g. Tc99m), might be produced compared with the Thick Target Yield (MBq/C/g) of, for example, metallic targets. For instance, the paper by Esposito et al., Evaluation of 99Mo and 99mTc productions based on a high-performance cyclotron, Sci. Technol. Nucl. Ins. 2013, 1-14, DOI: 10.1155/2013/972381, reports for the most interesting proton beam energies (15-20-25 MeV) the optimal target thickness (or the equivalent target mass-thickness) required to maximize the Tc99m production. Such considerations are not reported, instead, in such a work. The only parameter I see about the beam energy loss expected inside the powder-in-gas ring target, is the very few value (known as Particle Stopping Energy) reported in Table 1, i.e. 0.092 MeV. If correctly understood, that means the full energy of the proton beam (= hitting beam power) is stopped by the blades. Therefore is unclear how efficient such a system is comapared with the optimal target thickness of current solid targets.            

Moreover, I stress once again that, one of the main limiting problems I see for such a system, is the extremely high powder mass of the isotope-enriched material that would be needed, i.e. several tens/hundreds of grams. This is actually much higher than that required by the more traditional concepts of solid target (e.g. ~ 0.3 up to a very few grams). In addition, I would like the author to be clear that there would be no reason to produce a lot of Ci of short-lived (e.g. Tc99m) radionuclides if is then difficult to deliver it over a very large area, in the new, cyclotron-driven, production approach of local radioisotope, like e.g. the F18.  

Specific comments

An additional major revision is needed prior it might be accepted for publication. The revisions suggested, as well as specific comments, are reported directly in the Pdf file of the manuscript attached, using the commenting tools. They have to be considered by the author as a reference for the requested revision in manuscript drafting. Authors are asked to improve the manuscript in the highlighted parts and to meet the revisions requested or to make, in addition, their own modifications.

Author Response

Dear reviewer

Thank you for the comments and suggestions.

I have revised the document strongly and the supplementary documents is removed.

The Communication document is language edited by the MDPI services.

The document is changed into a Communication Document instead of an article. 

The design is changed in a powdery thick target set-up in the front with a nearly horizontal entering beam direction. The powder's mass required is hereby very small. The comparison and other claims are removed. Modeled and calculated data are presented now. 

The comments;

Point 1: 

The manuscript new version submitted by the author is now much more balanced. However, even with such a revised work, I remain doubtful about the effectiveness of such a powder-in-gas target system, with respect to the current solid target ones. 

Response 1: 

The study involves the conceptual design. The effectiveness might be more obvious because the design is changed in a 'frontal' powdery thick target instead of an angled beam concept. The behavior of the powder particles is strongly related, I expect, to the powder's prepeartion procedure which must be developed.

Point 2: 

The main focus of this work still remains the cinematics/dynamics parameters determination describing the behaviour of the spiralling carrier He gas approaching the steady-state, in a narrow ring-shaped cylindrical region where the accumulation of the isotope-enriched powder is supposed to occur. The same author at last concludes that, despite the promising design, some parameters have still to be investigated more in in depth. The main motivation given by the author is always the feasibility study about the very high heat power relief (in my opinion untrustworthy values) the syste would be able to manage, as also reported in Table 1 (e.g. the excessively high 100 kW beam power – at Which energy?) for Mo100(p,2n)Tc99m reaction, that I consider basically unpractical.

Response 2: 

The excessively high currents are now more in balance to the vendor numbers about cyclotron (future) capabilities. However, focusing on the Netherlands, about 5 to 6 cyclotrons would be needed using 300 μA beams. The report showed special attention to these nuclear reactions due to the broad range of interest and use. For the Netherlands, one or two cyclotrons running 2 mA daily 6 hours would cover the needs. Handling activation and contamination has to be controlled but every irradiation assembly becomes activated. The choice of materials and design is important. Comparing with our existing site at Eindhoven (Cygne center), daily production of multiple 1 TBq batches of I-123 and  F-18 on routine basis is possible by good material choices and almost seal-less assemblies. This Communication Document is related to the WTTC17 in Portugal. Experiments in near future requires resources which are not yet available. Conventional experimental work regarding the vortex principle is foreseen before the WTTC18 meeting.

Point 3: 

On the other hand, the information I did expected, about the resulted target mass thickness that such a system is able to manage, is still missing (e.g. for Mo100-enriched powders). Such a parameters is critical to better understand how, such a powder-in-gas target system, is far from a standard thick solid target (e.g. for Tc99m production). Therefore how much activity per unit mass and electric charge of a given radiosiotope (e.g. Tc99m), might be produced compared with the Thick Target Yield (MBq/C/g) of, for example, metallic targets. For instance, the paper by Esposito et al., Evaluation of 99Mo and 99mTc productions based on a high-performance cyclotron, Sci. Technol. Nucl. Ins. 2013, 1-14, DOI: 10.1155/2013/972381, reports for the most interesting proton beam energies (15-20-25 MeV) the optimal target thickness (or the equivalent target mass-thickness) required to maximize the Tc99m production. Such considerations are not reported, instead, in such a work. 

Response 3: 

The target mass is corresponding to the thick target (Zn-68 appr. 0.4mm and Mo-99 appr. 0.6mm for 14 and 20 MeV proton beam respectively). 

Point 4: 

The only parameter I see about the beam energy loss expected inside the powder-in-gas ring target, is the very few value (known as Particle Stopping Energy) reported in Table 1, i.e. 0.092 MeV. If correctly understood, that means the full energy of the proton beam (= hitting beam power) is stopped by the blades. Therefore is unclear how efficient such a system is comapared with the optimal target thickness of current solid targets. 

Response 4: 

The noted 0.092 MeV was the maximum energy uptake of one particle in the Bragg Peak range and not the total uptake by the powder. This was confusing because the document was unclear in that. 

Point 5: 

Moreover, I stress once again that, one of the main limiting problems I see for such a system, is the extremely high powder mass of the isotope-enriched material that would be needed, i.e. several tens/hundreds of grams. This is actually much higher than that required by the more traditional concepts of solid target (e.g. ~ 0.3 up to a very few grams). In addition, I would like the author to be clear that there would be no reason to produce a lot of Ci of short-lived (e.g. Tc99m) radionuclides if is then difficult to deliver it over a very large area, in the new, cyclotron-driven, production approach of local radioisotope, like e.g. the F18.   

Response 5:

The powders mass is now comparable with common solid target set-ups. This is due o the design change.

Point 6: 

Specific comments: An additional major revision is needed prior it might be accepted for publication. The revisions suggested, as well as specific comments, are reported directly in the Pdf file of the manuscript attached, using the commenting tools. They have to be considered by the author as a reference for the requested revision in manuscript drafting. Authors are asked to improve the manuscript in the highlighted parts and to meet the revisions requested or to make, in addition, their own modifications.

Response 6: 

The document is revised, shorter and language edited. 

All highlighted items in the pdf file where implemented first prior to the editing.

Thanks for this detailed reviewing,

Gerrie Lange

Reviewer 3 Report

The english in this document is still very poor. Important concepts are lost in what appears to be a Google translation. Getting the english right in such a complex description is imperative, since readers unfamiliar with the finer concepts of target design and beam dynamics will be lost trying to follow the manuscript.

It is difficult to suggest corrections with so many issues, but e.g. the sentence in line 31-33 makes little sense. I assume the author is talking about a flat top beam profile, it seems to quote a study that is not referenced and talks about a window that has no description (thickness, material etc.)

The document is getting better but a lot of work is still needed to get it publishable.

Author Response

Dear reviewer

Thank you for the comments and suggestions.

I have revised the document strongly. 

The document is language edited by the MDPI services.

The document is changed into a Communication Document instead of an article. 

The design is changed in a powdery thick target set-up. At the entrance with a nearly horizontal entering beam direction. The powder's mass required is hereby very small. The comparison and other claims are removed. Modeled and calculated data are presented now. 

The comments;

Point 1--2: Getting the English right in such a complex description is imperative.

Response 1--2: Due to the redesign of the chamber and beam direction, the design is much more obvious. The modeling work is established for ion energy ranges at the entrance to the threshold energy for specific nuclear reaction. The length of the powder layer corresponds to thick target in the model and calculations.

Document is checked and language edited by the mdpi services.

Thanks again,

Gerrie Lange

Round 3

Reviewer 2 Report

In this second revision step, the manuscript (now just a communication work instead of a research article), has been improved by the authors, also in the English language style, as requested. However further explanations in the highlighted parts, and additional modifications requests (misprint errors and/or /more basic errors etc.) are anyway necessary. As it is, it is not acceptable for publication yet. A further, minor, revision step is still needed, prior the manuscript might (eventually) be considered for a new submission.

The revisions suggested, as well as comments, are reported directly in the Pdf file of the manuscript attached, using commenting tool. They have to be considered for a reviewed manuscript drafting. The authors are asked to meet the revisions requested or to make, in addition, their own modifications aimed at further improving it.

Author Response

Dear reviewer

Thanks for all the comments which are carefully implemented.

The comments supported in the pdf-file with a yellow "button" are implemented by changes in the document.

I have them listed in the attachment file.

The document is a second time language edited by MDPI services.

Thanks,

Gerrie Lange 

Reviewer 3 Report

The author has in my mind done enough to shape the paper in a form for it to be published and be understood by the average reader.

Author Response

Dear reviewer

Thanks for all the comments so far and the acceptance.

Because of other reviewer comments, minor changes are implemented. 

The document is a second time language edited by MDPI services.

Thanks,

Gerrie Lange